# Ultra-High Contrast (UHC) MRI of the Brain, Spinal Cord and Optic Nerves in Multiple Sclerosis Using Directly Acquired and Synthetic Bipolar Filter (BLAIR) Images

**DOI:** 10.3390/diagnostics15030329

**Published:** 2025-01-30

**Authors:** Paul Condron, Daniel M. Cornfeld, Mark Bydder, Eryn E. Kwon, Karen Whitehead, Emanuele Pravatà, Helen Danesh-Meyer, Catherine Shi, Taylor C. Emsden, Gil Newburn, Miriam Scadeng, Samantha J. Holdsworth, Graeme M. Bydder

**Affiliations:** 1Mātai Medical Research Institute, Gisborne-Tairāwhiti 4010, New Zealand; p.condron@matai.org.nz (P.C.); g.newburn@matai.org.nz (G.N.);; 2Department of Anatomy and Medical Imaging and Centre for Brain Research, Faculty of Medical and Health Sciences, University of Auckland, Auckland 1010, New Zealand; 3Auckland Bioengineering Institute, Private Bag 92019, Auckland 1142, New Zealand; 4Multiple Sclerosis Society, Gisborne-Tairāwhiti 4010, New Zealand; 5Department of Neuroradiology, Neurocenter of Southern Switzerland, 6903 Lugano, Switzerland; 6Faculty of Biomedical Sciences, Universita della Svizzera Italiana, 6900 Lugano, Switzerland; 7Department of Ophthalmology, University of Auckland, Auckland 1010, New Zealand; 8Eye Institute, Auckland 1050, New Zealand; 9Department of Radiology, University of California San Diego (UCSD), San Diego, CA 92093, USA

**Keywords:** ultra-high contrast MRI, divided subtracted inversion recovery (dSIR) sequence, logarithmic then subtracted inversion recovery (lSIR) sequence, bipolar filter (BLAIR), multiple sclerosis, whiteout sign, grayout sign, white matter disease of the brain

## Abstract

In this educational review, the basic physics underlying the use of ultra-high contrast (UHC) bipolar filter (BLAIR) sequences, including divided subtracted inversion recovery (dSIR), is explained. These sequences can increase the contrast produced by small changes in T_1_ by a factor of ten or more compared with conventional IR sequences. In illustrative cases, the sequences were used in multiple sclerosis (MS) patients during relapse and remission and were compared with positionally matched conventional (T_2_-weighted spin echo, T_2_-FLAIR) images. Well-defined focal lesions were seen with dSIR sequences in areas where little or no change was seen with conventional sequences. In addition, widespread abnormalities affecting almost all of the white matter of the brain were seen during relapses when there were no corresponding abnormalities seen on conventional sequences (the whiteout sign). Grayout signs, in which there is a loss of contrast in gray matter or between gray matter and CSF, were also seen, as well as high signal boundaries around lesions. Disruption of the usual high signal boundary between white and gray matter was seen in leucocortical lesions. Lesions in the spinal cord were better seen or only seen with dSIR sequences. Generalized change was observed in the optic nerve with the dSIR sequence in a case of optic neuritis. UHC BLAIR sequences may be of considerable value for recognition of abnormalities in clinical practice and in research studies on MS.

## 1. Introduction

Ultra-high contrast (UHC) MRI is a term used to describe MRI which shows high-contrast demonstration of abnormalities in situations where little or no change from normal is seen on conventional state-of-the-art MR images. It is achieved without increase in static or gradient field strength.

With conventional MR sequences, change in a tissue property such as T_1_ is usually used only once to create contrast. With one form of UHC MRI, the same change in a tissue property is used three or four times in the same sequence to create much greater contrast [1]. An example of this is the divided subtracted inversion recovery (dSIR) sequence, where changes in T_1_ are used repeatedly to increase contrast. The contrast from small changes in T_1_ with dSIR sequences may be ten or more times greater than that obtained with conventional IR sequences such as MP-RAGE (magnetization prepared-rapid acquisition gradient echo). Using this approach, widespread changes have been seen in mild traumatic brain injury (mTBI) and hypoxic injury to the brain in regions which show no abnormality with state-of-the-art T_2_-weighted spin echo (T_2_-wSE) and/or T_2_-weighted fluid attenuated inversion recovery (T_2_-FLAIR) sequences [2,3,4]. This is likely to be because the changes in tissue properties such as T_1_ and T_2_ are too small to show abnormalities with conventional sequences, but produce obvious changes with the greater contrast amplification provided by dSIR sequences.

There is little point in demonstrating lesions that are already seen with high contrast using conventional sequences with even greater contrast, so the focus in applications of UHC MRI has been on areas of the brain that show little or no lesion contrast and appear normal with conventional sequences [5].

In an earlier paper, a single case of multiple sclerosis (MS) was described in order to demonstrate the technical capabilities of a dSIR sequence [1]. In this paper, we demonstrate more cases of MS imaged with dSIR sequences, as well as a new bipolar filter sequence, namely, log then subtracted inversion recovery (lSIR). The paper also illustrates how the T_1_-dependent dSIR and lSIR sequences can be supplemented by other tissue properties such as T_2_ in the form of composite bipolar filter sequences. MS patients were examined in remission and during relapse.

In the next section of this educational review, the basic physics underlying the use of bipolar sequences is described. This is followed by a comparison of conventional and bipolar sequences to determine where the new sequences are likely to be of clinical or research value.

## 2. Basic Physics

The first part of this section is a condensed and updated version of work published previously [1,5]. The second part describes new techniques.

### 2.1. Tissue Property Filters (TP-Filters) and the Inversion Recovery (IR) Sequence

Instead of illustrating image contrast in the usual way, by plotting longitudinal and then transverse magnetization (M_Z_ and then M_XY_) against time using the Bloch equations, it is possible to use plots of signals against T_1_ and T_2_, respectively. These plots are tissue property filters (TP-filters); the TPs may be mobile proton density, T_1_, T_2_, T_2_*, D*, etc.

For TR >> T_1_, the T_1_ dependent part of the IR sequence has the following signal equation:

S_T1_ = (1 − 2e^−TI/T^_1_)
(1)
 where S_T1_ is the signal from the T_1_-filter, TI in the inversion time, and T_1_ is the longitudinal magnetization relaxation time. Two T_1_-filters of the IR sequence with different values of TI (TI_s_ = TI_short_ and TI_i_ = TI_intermediate_) are shown in magnitude form in Figure 1A. The curves are negative unipolar filters.

The subtraction is as follows: first, TI_s_ T_1_-unipolar filter minus the second TI_i_ T_1_-unipolar filter produces the subtracted IR (SIR) T_1_-bipolar filter seen in Figure 1B. The vertical dashed lines at the null points of the two IR T_1_-filters seen in Figure 1 divide the X axes in Figure 1B,C into the lowest Domain (lD), the middle Domain (mD), and the highest Domain (hD). In the mD in Figure 1B (between the two dashed lines), the size of the slope of the SIR T_1_-bipolar filter is about double that of the IR T_1_-unipolar filters in the mD shown in Figure 1A. This is because the slope of the SIR filter in its mD in Figure 1B is the positive slope of the TI_s_ filter in the mD shown in Figure 1A minus the negative slope of the TI_i_ filter in the mD also shown in Figure 1A.

The two IR T_1_-filters shown in Figure 1A can also be added to give the Added IR (AIR) T_1_-filter shown in Figure 1C. In its mD, which is bounded by the vertical dashed lines, the signal is reduced to about 0.20 compared with its value of 2 at T_1_ = 0 (i.e., about one tenth of the value).

Figure 2A shows a divided SIR (dSIR) T_1_-bipolar filter in which the SIR T_1_-bipolar filter in Figure 1B is divided by the AIR T_1_-filter in Figure 1C, i.e., $\frac{\mathrm{difference}}{\mathrm{sum}}$. The signal S_dSIR_ of the dSIR filter is given by the following:(2)$$S_{\mathrm{dSIR}}=\frac{S_{\mathrm{TIs}}-S_{\mathrm{TIi}}}{S_{\mathrm{TIs}}+S_{\mathrm{TIi}}}$$
where S_TIs_ is the signal from the shorter TI filter and S_TIi_ is the signal from the intermediate TI_i_ filter.

The resulting dSIR T_1_-bipolar filter (Figure 2A) shows a negative pole and a positive pole. Its mD shows a highly positive nearly linear slope whose magnitude is about ten times greater than the slopes of the IR T_1_-filters shown in Figure 1A. The dSIR filter has maximum and minimum signal values of ±1. (The contributions from mobile proton density and T_2_ to conventional IR sequences cancel out with dSIR sequences, which are, therefore, T_1_ maps. As a result, the whole sequence can be accurately represented by its T_1_-bipolar filter.)

Figure 2B compares the contrast generated by a conventional S_TIs_ IR T_1_-unipolar filter (pink) to that from the SIR T_1_-bipolar filter (blue) shown in Figure 1B, from the same increase in T_1_ (ΔT_1_) (horizontal positive green arrow). Using the small change approximation of differential calculus, ΔT_1_ is multiplied by the slopes of the respective S_TI_ IR and SIR filters (red lines) to produce the differences in signal ΔS shown, i.e., the contrast resulting from each of them. This is given by the vertical pink and blue arrows on the right. The SIR T_1_-bipolar filter generates about twice the contrast (blue arrow) of the S_TIs_ IR T_1_-unipolar filter (pink arrow) from the same increase in T_1_, ΔT_1_. The S_TIs_ IR T_1_-filter is that of a conventional IR sequence such as MP-RAGE.

Figure 2C compares the contrast produced by the S_TIs_ IR T_1_-unipolar filter (pink) to that produced by the dSIR T_1_-bipolar filter shown in Figure 2A (blue). The increase in T_1_ ΔT_1_ (horizontal positive green arrow) is multiplied by the slopes of the respective S_TIs_ IR and dSIR filters (red lines) to produce the differences in signal ΔS shown, i.e., the contrast generated by the two filters. This is given by the vertical pink and blue arrows on the right. From the same increase in T_1_ (ΔT_1_), the dSIR T_1_-bipolar filter produces contrast about ten times greater than the contrast produced by the S_TIs_ IR T_1_-unipolar filter.

To produce the large increase in contrast shown in Figure 2C, the dSIR sequence is typically targeted at small increases in the T_1_ of normal tissue produced by disease (shown by the horizontal green arrow). These increases are positioned within the steeply sloping mD of the dSIR T_1_-bipolar filter. This is achieved by choosing appropriate values of TI_s_ and TI_i_. The target is often small increases in the T_1_ of white matter.

High contrast can also be produced by difference or change in T_1_ within the lower part of the hD and the upper part of the lD, where the dSIR T_1_-bipolar filter has relatively steep slopes. (The term ‘tissue’ is used to include fluids in the rest of this paper unless otherwise specified).

### 2.2. Contrast at Tissue Boundaries

MRI tissue boundaries take different forms, such as a gradual change in signal from one tissue to another, as well as sharply defined high and low signal boundaries between tissues.

In the boundary region between two pure tissues (such as between white and gray matter), the T_1_s of voxels which contain mixtures of the two tissues typically span the range of T_1_ values between those of the two pure tissues. If a narrow mD dSIR T_1_-bipolar filter (e.g., with TI_s_ nulling normal white matter and TI_i_ longer than TI_s_ but less than that needed to null gray matter) is used, there are mixtures of the two tissues in voxels in the boundary region, with intermediate T_1_ values (T_1W,G_) which correspond with the peak signal of the T_1_-bipolar filter (S_W,G_), as shown in Figure 3 (vertical blue arrow). This produces a high signal boundary between white matter and gray matter. The dSIR T_1_-bipolar filter shows much higher contrast than that produced between white matter and gray matter by a conventional white matter nulled IR sequence such as MP-RAGE (Figure 3) (vertical pink arrow). High contrast boundaries are also seen at the junction between white matter and CSF at ventricular boundaries through the same mechanism.

### 2.3. T_1_ Maps and Qualitative—Quantitative MRI

To better understand the dSIR T_1_-bipolar filter, a linear equation of the form y = mx + c can be used to approximate the filter in its mD. The equation is produced by fitting a straight line between the first and last points of the mD (i.e., first point x = $\frac{{\mathrm{TI}}_{s}}{\ln2}$ and y = −1, and last point x = $\frac{{\mathrm{TI}}_{i}}{\ln2}$ and y = +1). In the mD, the signal of the dSIR sequence S_dSIR_, is given by the following:(3)$$S_{\mathrm{dSIR}}\;\approx \;\frac{\ln\;4}{∆TI}\;\;T_{1}-\frac{\Sigma TI}{∆TI}$$
where ΔTI = TI_i_ − TI_s_ (i.e., second TI minus first TI), which is positive, and ΣTI = TI_s_ + TI_i_, which is also positive. Note that, because ΔTI is positive, the slope $\frac{\ln\;4}{∆TI}$ is positive. The offset is negative. T_1_ is the longitudinal magnetization relaxation time.

The expression in Equation (3) illustrates four key features of the dSIR T_1_-bipolar filter, firstly, the near linear change in signal (i.e., S_dSIR_) with T_1_ in the mD, secondly, the filter has a slope in the mD equal to $\frac{\ln4}{\Delta \mathrm{TI}}$, thirdly, the filter shows a high sensitivity to small changes in T_1_ when the size of ΔTI is small. When ΔT_1_ is small, the size of ΔTI can be decreased to match it, and so the sensitivity of the filter is scaled up. This is because the reduction in ΔTI increases the steepness of the T_1_-filter in the mD and, thus, the amplification of the contrast produced from ΔT_1_. It compensates for the small value of ΔT_1_ until the sequence becomes SNR and/or artefact limited. This is not the case with conventional IR sequences, where, if ΔT_1_ is decreased, contrast is decreased.

Fourthly, Equation (3) can be used to calculate T_1_ values in the mD, as follows:(4)$$T_{1}\approx \frac{∆TI}{ln4}\;S_{\mathrm{dSIR}}+\frac{\Sigma TI}{ln4}$$

The terminology is the same as for Equation (3).

The linear approximation is only valid in the mD.

T_1_s in the mD occupy the full range of the display gray scale from black to white. Outside of the mD, in the lD and hD, the magnitude of the T_1_-bipolar filter signal decreases towards zero at minimum and maximum values of T_1_. T_1_s, from the shortest and longest T_1_s in the lD and hD, respectively, only occupy parts of the display gray scale.

### 2.4. Log then Subtracted Inversion Recovery (lSIR) Sequences

Another bipolar filter is the natural logarithm (ln) then subtracted inversion recovery (lSIR) filter. This is half of the following subtraction: ln short S_TIs_ minus ln intermediate S_TIi_. Figure 4 shows a dSIR T_1_-bipolar filter in blue and the corresponding lSIR T_1_-bipolar filter with the same TI_s_ and TI_i_ in orange. The two filters appear similar in the lowest region of the lD and in the highest region of the hD, as well as in the central region of the mD. The slope of the dSIR filter (blue) in the mD is essentially constant and equal to $\frac{\ln4}{\Delta \mathrm{TI}}$. The slope of the lSIR filter (orange) is the same as that of the dSIR filter at the center of the mD, but greater around the lower and higher nulling TI values. The signal of the lSIR filter asymptotically approaches negative and positive infinity at the two corresponding nulling T_1_s. The lSIR filter increases its slope, and, thus, its contrast amplification, as the change in T_1_ from the nulling value of T_1_ becomes smaller. This compensates for the decrease in the size of the change in T_1_. The low and high signal peaks of the lSIR filter are narrower than those of the corresponding dSIR filter, which results in narrower boundaries between white and gray matter.

Like the dSIR filter, the lSIR filter essentially eliminates the mobile proton density and T_2_ dependence of the full IR sequence and so can be used to model the contrast behavior of the full IR sequence.

From a practical point of view, the lSIR T_1_-filter has 2–3 times the slope of the dSIR T_1_-filter for very small differences or changes in T_1_ around the null points, and, thus, has a 20–30 times greater contrast than conventional IR sequences for the same very small differences or changes in T_1_. The maximum and minimum values of the lSIR filter are usually shown as ±2–3 (rather than ±infinity). This compares with values of ±1 for the dSIR filter.

The lSIR filter signal, S_lSIR_, is the inverse hyperbolic tangent of the dSIR filter signal, S_dSIR_, in the mD (S_lSIR_ = atanh S_dSIR_) and the inverse hyperbolic cotangent of the dSIR filter signal in the lD and hD.

### 2.5. Composite (c) Bipolar Filters (T_1_ as well as T_2_, T_2_*, and/or D*)

It is possible to introduce an attenuation filter S_OF_ (S_other filter_) in the form of an additional segment of the IR pulse sequence and apply it to one of the two IR sequences used for dSIR T_1_-bipolar filter imaging in Figure 1 and Figure 2. S_OF_ may equal e^−ΔTE/T2^, e^−ΔTE/T2*^, or e^−ΔbD*^ (where ΔTE and Δb are the differences in TE or b [the diffusion sensitivity parameter] between the two IR sequences). This, respectively, introduces T_2_, T_2_*, and D* dependence. It provides additional contrast to the T_1_ contrast of the dSIR T_1_-bipolar filter. When ΔTE equals zero, or Δb equals zero, S_OF_ = 1 and there is no attenuation. If S_OF_ is set to 1.0, 0.5, and 0.25, the curves shown in Figure 5A,B result when using a narrow mD dSIR sequence (TI_s_ = 350 ms and TI_i_ = 500 ms). The nulling times are the same for all values of S_OF_.

In Figure 5A, attenuation of the first IR filter with TI_s_ is shown with S_OF_ set at 1 (blue), 0.5 (orange), and 0.25 (yellow). The yellow and orange curves are wider than the unattenuated dSIR curve (blue) and are seen around the outside of the first (negative) pole of the blue curve at the first TI_s_. At the second positive pole, the yellow and orange curves are narrower than the unattenuated blue curve and are seen inside it. This results in a broadening, loss of contrast, and lower signal white matter around the first (negative) pole, as well as increased contrast and narrower boundaries between white and gray matter around the second (positive) pole.

In (A), as S_OF_ is decreased from 1.0 to 0.5 and 0.25, the two latter filters become wider around the first (negative) pole and are positioned outside the blue filter. At the second (positive) pole, the attenuated filters are inside the blue filter. They show higher slopes.

(B) shows the conventional T_1_-bipolar filter (blue) with S_OF_ = 1, an attenuated TI_s_ filter with S_OF_ = 0.5 (red), and a further attenuated TI_s_ filter with S_OF_ = 0.25 (yellow). As S_OF_ is attenuated, the S_OF_ = 0.5 and S_OF_ = 0.25 filters around the first (negative) pole are situated inside the blue curve, become narrower, and have higher slopes. The attenuated filters around the second (positive) pole are situated outside the blue curve. They are wider and show lower slopes. For T_2_, a short T_2_ increases attenuation relative to a longer T_2_, and this corresponds to the narrower and sharper peak, with greater negative contrast and spatial resolution of the contrast in (A).

In Figure 5B, attenuation of the second IR filter with TI_i_ is shown. As S_OF_ is decreased to 0.5 and 0.25, the filter narrows around the inside of the blue filter at the first (negative) pole and widens around the outside of the blue filter at the second (positive) pole. The negative contrast produced for the same difference in T_1_ is increased around the first pole and decreased around the second pole.

The same principles can be applied to the lSIR filter. Composite (c) forms of the bipolar filters can be designated as cSIR, cdSIR, clSIR, etc.

Susceptibility and magnetization transfer (MT) can also be used to attenuate signals and produce supplementary contrast. The combination of T_1_ and T_2_* contrast may be of particular value in demonstrating the paramagnetic rim sign and the central vein sign in MS.

## 3. Methods

With approval from the New Zealand Health and Disability Ethics Committee (EXP 11360, 2022), and informed consent from each subject, MR scans were performed on four adult normal controls and nine adult patients with MS during relapse and remission or with suspected MS (one case). A 3T scanner (General Electric Healthcare Premier) was used. Two-dimensional IR fast spin echo (FSE) sequences were performed with a TIs nulling normal white matter, and a longer TIi was chosen to produce narrow mD dSIR images targeted at small increases in the T_1_ of white matter (Figure 2C). Two-dimensional and three-dimensional wide mD dSIR images with a short TI_s_ and a long TI_l_ were also obtained. Synthetic narrow mD dSIR images were created from wide mD dSIR images [1], and dSIR images were used to synthetically produce lSIR images. Positionally matched 2D and 3D T_2_-wSE and/or T_2_-FLAIR images were acquired, as well as filtered susceptibility weighted images (Table 1).

## 4. Illustrative Cases

Figure 6 shows mid-ventricular images of a 41-year-old female patient with MS in remission. No abnormality is seen on the T_2_-wSE image (Figure 6A). On the narrow mD dSIR image (Figure 6B), normal peripheral white matter is dark and has a low signal (thick arrows). The more central superior longitudinal fasciculi show a higher signal with a gradient of signal decreasing from posterior to anterior. Within them are the corticospinal tracts (short thin arrows). Three focal lesions are seen in the peripheral white matter and at the leucocortical junctions (long thin arrows).

At a supraventricular level in the same patient (Figure 7), the T_2_-wSE image is again normal (Figure 7A). The corresponding narrow mD dSIR image shows only a small region of low signal normal white matter (dark arrow). The corticospinal tracts are seen (short thin white arrows), as well as a lesion (long thin white arrow). There are widespread asymmetric and patchy abnormalities in the white matter.

Figure 8 shows positionally matched supraventricular T_2_-FLAIR (Figure 8A) and narrow mD dSIR (Figure 8B) images in a patient with MS during a relapse. A single lesion is seen on the T_2_-FLAIR image, and this is also seen on the dSIR image (long white arrows). An additional six lesions are only seen on the dSIR image (short white arrows). Many of the lesions on the dSIR image have high signal boundaries (rims). In addition, there are bilateral symmetrical widespread relatively uniform increases in signal in white matter, with some sparing of more peripheral white matter. These are features of a high grade (4–5) (maximum grade 5) whiteout sign [6]. This differs from the normal dark appearance of white matter seen on the narrow mD dSIR image in Figure 6B. No evidence of a whiteout sign is seen on the T_2_-FLAIR image.

Figure 9 and Figure 10 show a 38-year-old patient with MS in remission (Figure 9A and Figure 10A, left columns) and during a relapse two years and five months later (Figure 9B and Figure 10B, right columns) imaged with the same narrow mD dSIR sequence. Figure 9 shows three positionally matched lower levels in the brain and Figure 10 shows two positionally matched higher levels in the brain. White matter shows a generally low signal during remission (whiteout sign grade 1–2) (Figure 9A and Figure 10A), and a high signal (whiteout sign grade 4–5) during the relapse (Figure 9B and Figure 10B). The multiple levels within the brain show the wide distribution of the high grade whiteout sign in the cerebellar and cerebral hemispheres, as well as in the brainstem. No evidence of a whiteout sign was seen on the corresponding T_2_-FLAIR images.

Figure 11 shows narrow mD dSIR images in a normal adult control (Figure 11A) and in a 77-year-old patient with MS (Figure 11B) during a relapse. In addition to the whiteout sign, there is a loss of contrast between gray and white matter in the thalamus of the patient (Figure 11B). Also, the heads of the caudate nuclei, as well as the insular and peripheral cortices, appear isointense with CSF on the dSIR image. These are grayout signs. No evidence of a whiteout sign or grayout signs was seen on the corresponding T_2_-FLAIR images.

Figure 12 shows a 41-year-old female patient with MS during a relapse. Wide mD dSIR (Figure 12A) and phase filtered susceptibility weighted gradient echo (Figure 12B) images are shown. The dSIR image shows well defined high signal boundaries around two lesions (Figure 12A, white arrows). These lesions show less obvious paramagnetic rim signs on the filtered susceptibility weighted images (Figure 12B, white arrows). The high signal in Figure 11A could be due to increased T_1_ in the lesions beyond the second T_1_ nulling value and/or reduction in T_1_ in long T_1_ lesions due to free iron.

Figure 13 shows a leucocortical lesion in a 41-year-old female patient with MS in remission. The lesion appears blurred and relatively featureless on the narrow mD dSIR image (Figure 13A), but shows a disrupted boundary within the lesion on the lSIR image (Figure 13B). Figure 13C shows dSIR (blue) and lSIR (orange) signal profiles across the lesion. These are plots of signal vs. distance in mms on the image. The lSIR profile has higher signal and generally steeper slopes than the dSIR profile, corresponding to the graphs in Figure 4.

Figure 14 shows another leucocortical lesion in a 24-year-old female patient with MS in remission. dSIR (Figure 14A) and cdSIR (Figure 14B) images are compared. The cdSIR image shows higher contrast. In addition, the frontal white matter has a more uniform low signal intensity on the cdSIR image. These are features consistent with the graphs in Figure 5A.

Figure 15 shows T_2_-FLAIR (Figure 15A) and wide mD dSIR (Figure 15B) sagittal images of the upper cervical spinal cord in a 43-year-old female patient with MS in remission. The T_2_-FLAIR image shows a poorly defined smudge of increased signal (white arrow). In the corresponding position, the dSIR image (Figure 15B) shows a high-contrast lesion with sharply defined boundaries, which is much more extensive than in (Figure 15A) (lower three arrows). An additional lesion is seen in the medulla on the dSIR image (highest arrow), but is not seen on the T_2_-FLAIR image.

Figure 16 shows the upper cervical spinal cord in a 42-year-old female patient with MS in remission. T_2_-FLAIR (Figure 16A), T_2_-wSE (Figure 16B), and wide mD dSIR (Figure 16C) images are compared. A large lesion is seen on all three images, but with greatest contrast on the dSIR image (long white arrows). In addition, three small lesions are only seen on the dSIR image (Figure 16C, short white arrows).

In a patient with optic neuritis and suspected MS, Figure 17A,B (upper row) show axial fat saturated T_2_-FLAIR (Figure 17A) and narrow mD dSIR (Figure 17B) images of their optic nerves. Figure 17C,D (lower row) show parasagittal oblique fat saturated T_2_-FLAIR (Figure 17C) and narrow mD dSIR (Figure 17D) images of the left optic nerve. The right optic nerve appears normal in Figure 17A,B. The left optic nerve shows a distal abnormality on the T_2_-FLAIR axial and parasagittal images (white arrows), but shows proximal and distal abnormalities on the corresponding dSIR images (white arrows).

## 5. Discussion

Use of bipolar filter sequences such as dSIR, lSIR, and cdSIR demonstrated abnormal appearances in the brain, spinal cord, and optic nerve in MS patients which were not shown, or were only poorly shown, with T_2_-wSE and T_2_-FLAIR sequences. These included focal lesions, the whiteout sign, grayout signs, high signal boundaries/rims around lesions, disruption of boundaries between white and gray matter, and generalized changes in the optic nerve.

Unipolar filter IR sequences such as intermediate TI IR, STIR, MP-RAGE, and MP2RAGE have been employed in clinical MRI for many years, but bipolar filter sequences (Table 2) are a new development [1,4]. Detail of the mathematics describing them is included in Table 3. The equations also provide a basis for implementing synthetic bipolar sequences which can be produced from existing T_1_ sequences or a combination of TP maps and synthetic bipolar filters.

UHC MRI with bipolar filter sequences uses IR sequences which already exist on most clinical MR systems and require minimal image processing. They can be implemented in a short time at very low cost and can be used at any clinical static field.

### 5.1. T_1_ Measurements and Magnetization Transfer (MT)

The dSIR sequence is a T_1_ map with a linear relationship between S_dSIR_ and T_1_ in the mD (Equation (4)). T_1_ measurements obtained with dSIR sequences have been validated in phantom and human studies [7,8]. The observed values of T_1_ (T_1obs_) are generally shortened by incidental MT effects arising from off-resonance 180° pulses when FSE data collections are used. MT results in a decrease in observed mobile proton density and a proportionate decrease in T_1obs_. As a general rule, in diseased tissue compared with normal tissue, MT effects are decreased relative to normal. This results in a relative increase in T_1_ in diseased tissue, which is synergistic with increases in T_1_ arising in disease from other causes.

### 5.2. Signs

#### 5.2.1. Whiteout Sign

The whiteout sign consists of a bilateral, symmetrical, widespread, generally uniform increase in signal in the white matter of the cerebral and cerebellar hemispheres, as well as the brainstem. The anterior and posterior central corpus callosum, as well as peripheral white matter of the cerebral hemispheres are usually relatively spared. Whiteout signs can be graded from one (normal) to five (abnormal) [6]. They are reversible and have been seen in mild traumatic brain injury (mTBI), hypoxic injury to the brain, cerebral tumors, long COVID, and MS. Whiteout signs are generally associated with neuroinflammation, although edema, demyelination, and degenerative change may also be implicated. When seen with the dSIR sequence nulled for normal white matter, they are due to widespread small increases in the T_1_ of white matter which are insufficient to generate visible contrast with conventional clinical sequences.

#### 5.2.2. Grayout Signs

Grayout signs can be manifest as a reduction or loss of contrast within a gray matter structure such as the thalamus, as well as reduction or loss of contrast between gray matter and white matter or between gray matter and CSF. On narrow mD dSIR sequences, a grayout sign can be caused by an increase in T_1_ in gray matter within the hD. Not infrequently, grayout signs are seen with higher grade whiteout signs, and, thus are features of an encephalopathy rather than a leucoencephalopathy (when only white matter is involved).

High signal boundaries and lesions with narrow mD dSIR sequences can arise with an increase in T_1_ in the lesion sufficient to cross the peak filter value and enter the hD. This corresponds with the dark rim around MS lesions seen with gray matter nulled double IR sequences, which occurs when there are relatively large increases in T_1_. This finding was seen more frequently in MS than in other diseases [9].

The high signal boundary/rim can also arise from a decrease in T_1_ in an MS lesion induced by free iron from myelin at the edge of an MS lesion. The decrease in T_1_ is from the hD back to the mD and passes through the peak of the bipolar filter.

High signal boundaries around lesions may be less obvious in the presence of a whiteout sign because of the widespread high signal in white matter, although many are apparent in Figure 8B.

### 5.3. Clinical Issues

#### 5.3.1. Activity

The usual signs of activity in MS are an increase in number and/or size with T_2_-weighted imaging in repeat studies, as well as gadolinium based contrast agent (GBCA) enhancement. To this, it may be possible to add the presence of a whiteout sign or increase in size or extent of a whiteout sign.

By using drSIR sequences sensitized to decreases in T_1_, it may be possible to increase the sensitivity of conventional IR sequences to GBCA enhancement by an order of magnitude, and, thus, increase the sensitivity of MRI to disease activity [1].

#### 5.3.2. Acute Clinical Episodes with Stable MRI (ACES)

Clinical episodes without changes in MRI appearances were recorded in 26.1% of episodes in a real-world study of MS [10]. Clinical radiological mismatch in MS during relapse has also been described in other papers [11,12]. There is a need for greater MRI sensitivity to correlate better with clinical features. This could be provided by bipolar filter sequences.

Smoldering lesions are recognized by progressive increase in size of abnormalities on T_2_-weighted images and paramagnetic rim lesions without GBCA enhancement [13,14,15,16,17], and could benefit from more sensitive sequences. The same applies to the use of T_2_-FLAIR imaging and GBCA enhancement to assess therapeutic response. Paramagnetic lesion rims [18,19,20,21] and the central vein sign [22,23,24,25,26] could also benefit from increased sensitivity.

#### 5.3.3. Diagnostic Criteria for MS

The core sequences for recognition of MS lesions are T_2_-wSE and/or T_2_-FLAIR sequences [27,28,29,30], and thus are based on changes in T_2_, not the changes in T_1_ seen with dSIR sequences. The sensitivity of dSIR sequences to the presence of MS lesions is supported by the belief that increases in T_2_ are accompanied by increases in T_1_, and so detection of either may be of similar clinical significance. The initial study of MS with MRI was performed with a T_1_-weighted IR sequence.

Conventional diagnostic criteria for MS treat patchy changes due to increased white matter signal seen with T_2_-weighted sequences as of no diagnostic significance. These features may be much more obvious with dSIR sequences (Figure 7). In addition, whiteout signs may be seen in a characteristic pattern throughout the white matter of the brain.

#### 5.3.4. Clinical Use

In clinical practice, it is possible to positionally match T_2_-FLAIR sequences with dSIR T_1_-bipolar filter (BLAIR) sequences. Larger changes in T_2_ are shown by T_2_-FLAIR sequences and smaller changes in T_1_ by T_1_-BLAIR sequences, which are, thus, complementary, as seen in Figure 8. cdSIR, T_1_, T_2_-BLAIR sequences can be used to add T_2_-weighting to the T_1_-weighting of dSIR sequences.

### 5.4. Other Sequences Using Two IR Sequences

Two other sequences employing two IR sequences are the nearest comparison to the BLAIR sequences described in this paper.

The first is the MP2RAGE sequence which uses two IR sequences with different TIs [31]. These signals are multiplied together and divided by the sum of their squares. This results in a negative unipolar filter with slope for white matter greater than that of the original IR sequences [32,33].

The second is the FLAWS (fluid and white matter suppression) group of sequences, which also uses two IR sequences with different TIs [32,33] that are subtracted and divided by their sum. Compared with narrow mD dSIR sequences, the ΔTIs of FLAWS high contrast (FLAWShc) and FLAWS high contrast opposite (FLAWShco) are wide and the sequences are monotonic, not bipolar, because they use signed or phase-sensitive signal values, not magnitude values.

The FLAWSmin sequence takes the minimum value of the two IR sequences and produces a positive unipolar filter with a wide ΔTI. It has shown halo signs of increased signal at the margins of MS lesions in gray matter or at the leucocortical boundary [34,35]. It has similarities to the dSIR sequence.

### 5.5. Further Developments

Further developments include 2D and 3D registration of images with different TIs as part of the basic sequence, but also registration of images before and after GBCA administration and for serial studies at different examinations.

The T_1_, T_2_*-BLAIR sequence is being developed to produce synergistic contrast from paramagnetic effects due to free iron.

Longer TI values are being used to null blood, and so visualize synergistic T_1_ and T_2_* effects due to central veins within MS lesions.

Particular variants of sequences designated to specifically see white matter changes in the spinal cord are being assessed together with narrow mD sequences with longer TIs for the thalamus.

It is possible to visualize lesions in the cortex using a longer second TI of 900 ms or 950 ms so that the high signal boundary is between the cortex and CSF not between white and gray matter. Boundaries can also be narrowed by the use of lSIR sequences rather than dSIR sequences.

## 6. Conclusions

Bipolar filter sequences promise a variety of options which appear well suited to imaging changes in the brain in MS. A program of technical development is in place to expand the range of available sequences and improve their effectiveness in diagnosis. Detailed clinical evaluation will be required. UHC MRI with bipolar filters could provide a substantial advance in the imaging of MS in clinical and research studies with MRI.

## Figures and Tables

**Figure 1 diagnostics-15-00329-f001:**
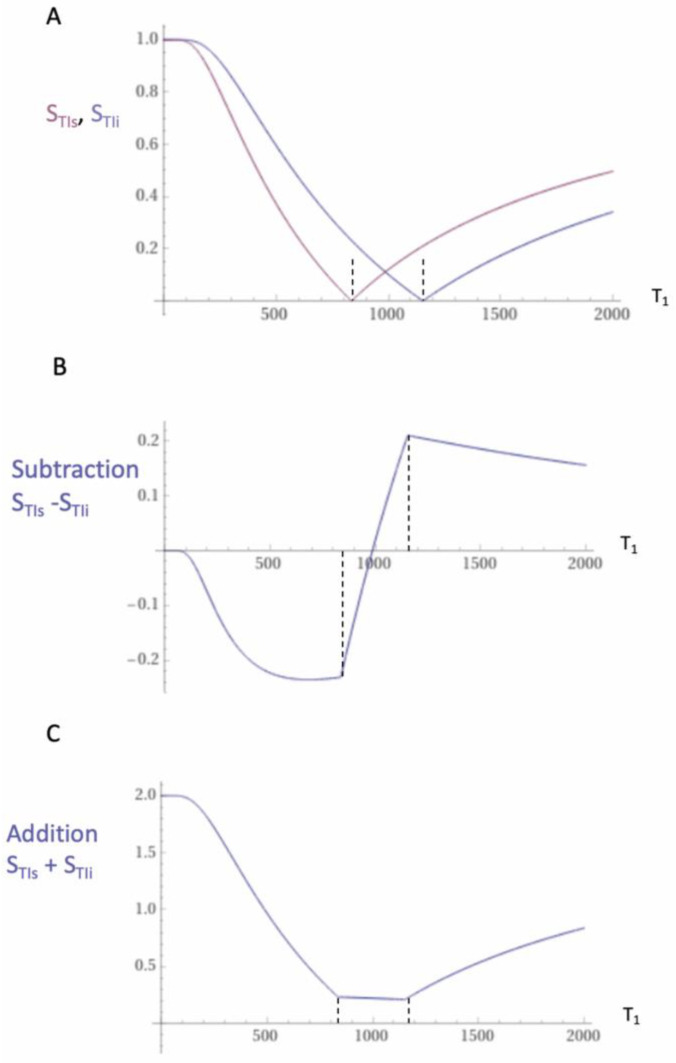
Unipolar IR (**A**), subtracted IR (SIR) (**B**), and Added IR (AIR) (**C**) T_1_-filters. T_1_ is shown along the X axes in ms. (**A**) shows a TI_s_ T_1_-unipolar filter (pink) and a TI_i_ T_1_-unipolar filter (blue), (**B**) shows the subtraction (S_TIs_ − S_TIi_) IR or SIR T_1_-bipolar filter, and (**C**) shows the addition (S_TIs_ + S_TIi_) IR or AIR T_1_-filter. The vertical dashed lines divide the X axes into lD, mD, and hD segments with the mD central. In (**B**), the slope of the curve in the mD of the SIR T_1_-filter is about double that of the S_TIs_ T_1_-filter (pink in [**A**]). In (**C**), the signal at T_1_ = 0 is doubled to 2.0, and the signal in the mD is about 0.20 in the central part of the AIR T_1_-filter (i.e., in the mD).

**Figure 2 diagnostics-15-00329-f002:**
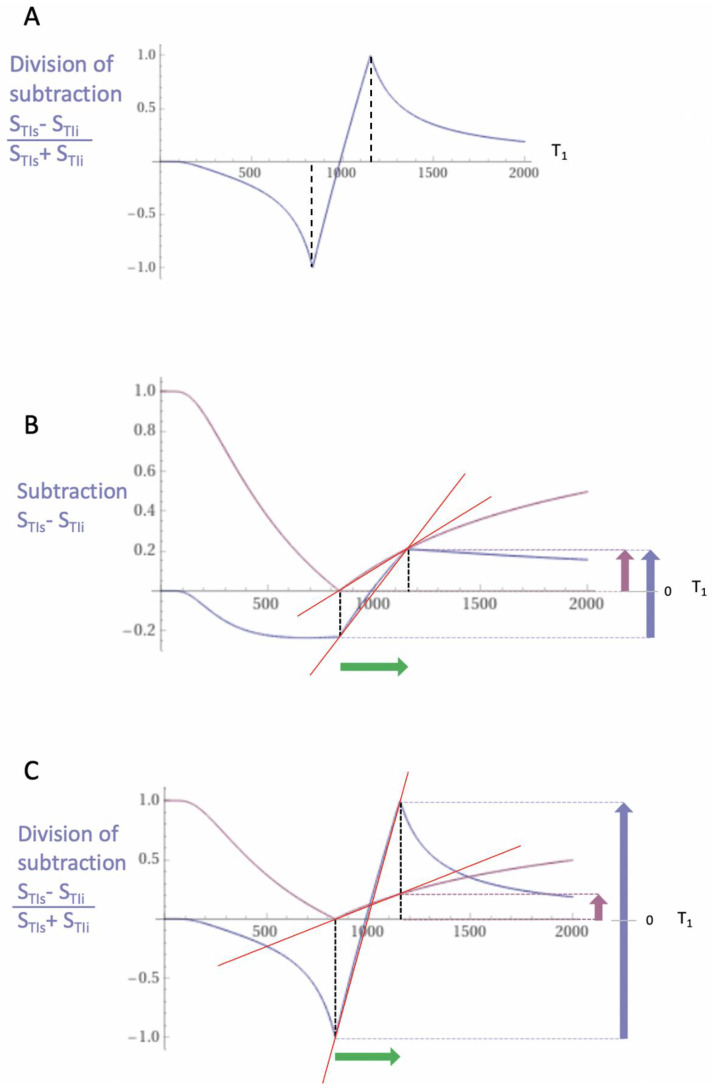
(**A**) shows division of the SIR T_1_-bipolar filter (Figure 1B) by the addition AIR T_1_-filter (Figure 1C) to give the dSIR T_1_-bipolar filter. (**B**) shows comparison of a conventional IR S_TIs_ T_1_-unipolar filter (pink) with the SIR T_1_-bipolar filter (blue) for a small increase in T_1_ (ΔT1) (horizontal green arrow). ΔTI is multiplied by the slopes of the filters to give the differences in signal or contrast, ΔS. The dSIR T1-bipolar filter produces about twice the contrast of the STIs T1-unipolar filter for the SIR filter (blue and pink arrows). (**C**) compares the S_TIs_ T_1_-unipolar filter (pink) with the dSIR T_1_-bipolar filter (blue) for the same small increase in T_1._ The dSIR T_1_-bipolar filter produces about ten times more contrast than the S_TIs_ T_1_-unipolar filter (blue and pink arrows).

**Figure 3 diagnostics-15-00329-f003:**
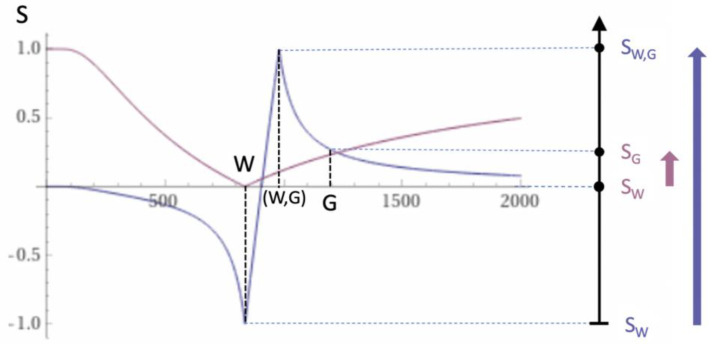
Contrast at tissue boundaries. This image shows a narrow mD dSIR T_1_-bipolar filter with a mD extending from the T_1_ of white matter (W) to a T_1W,G_ between the TIs of W and gray matter (G) (blue curve), as well as a white-matter-nulled conventional IR T_1_-unipolar filter e.g., MP-RAGE (pink curve). The X axis is shown in ms. The contrast produced by the T_1_-unipolar filter from the difference in signal between W and G is (S_G_ minus S_W_) and is shown by the vertical pink arrow. With the dSIR T_1_-bipolar filter, there is a partial volume effect between W and G, producing a T_1W,G_ between the T_1_s of W and G, which results in the high signal S_W,G_ shown on the T_1_-bipolar filter (blue curve). The contrast between this high signal and white matter is the difference (S_W,G_ minus S_W_, in blue) and is shown by the vertical blue arrow. The contrast produced by the T_1_-bipolar filter at the boundary between W and G is far greater than that produced at the boundary between W and G by the T_1_-unipolar filter.

**Figure 4 diagnostics-15-00329-f004:**
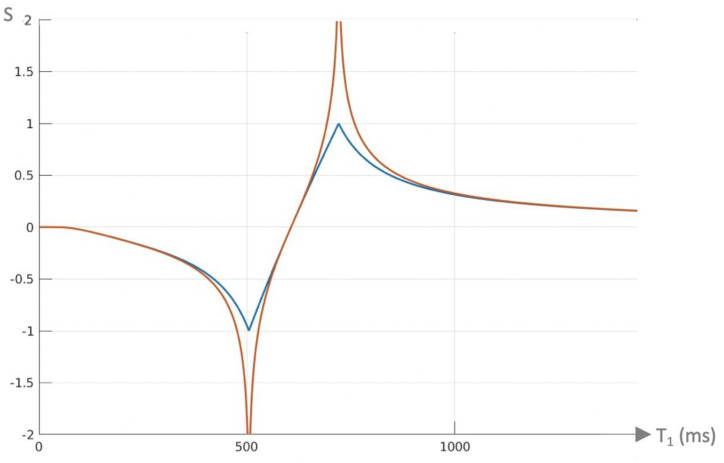
Plot of signal vs. T_1_ in ms for a dSIR T_1_-bipolar filter (blue curve) and the corresponding lSIR T_1_-bipolar filter with the same nulling points (orange curve). The dSIR filter shows the pattern seen in Figure 2 and Figure 3 with maximum values of ±1. The lSIR filter has similar values to dSIR filter in the regions of lowest and highest values of T_1_, as well as in the center of the mD. However, around the lower and upper T_1_ nulling values it is much steeper and proceeds asymptotically to minus and plus infinity respectively (values along the Y axis in the range of ± 2 are shown). For the same difference in T_1_ in boundary regions, the lSIR filter is steeper than the dSIR filter. This results in greater contrast. The contrast for very small differences or changes in T_1_ is typically about 2–3 times greater with the lSIR sequence compared with the corresponding dSIR sequence.

**Figure 5 diagnostics-15-00329-f005:**
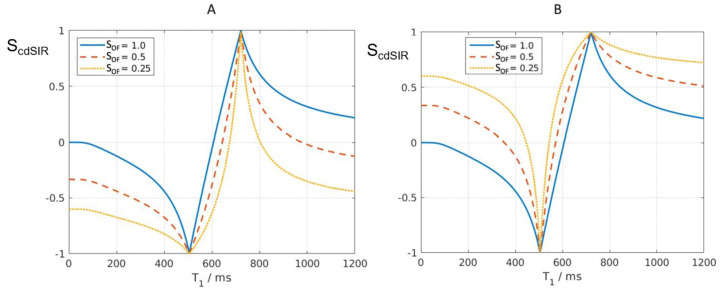
Composite dSIR filters (cdSIRs) with TI_s_ = 350 ms and TI_i_ = 500 ms. Attenuated first IR T_1_-filter (TI_s_) in (**A**) and attenuated second IR T_1_-filter (TI_i_) in (**B**). (**A**,**B**) show the conventional unattenuated T_1_-bipolar filter (blue) with S_OF_ = 1, an attenuated first TI_s_ filter (S_OF_ = 0.5) (red) and a further attenuated first TI_s_ filter (S_OF_ = 0.25) (yellow).

**Figure 6 diagnostics-15-00329-f006:**
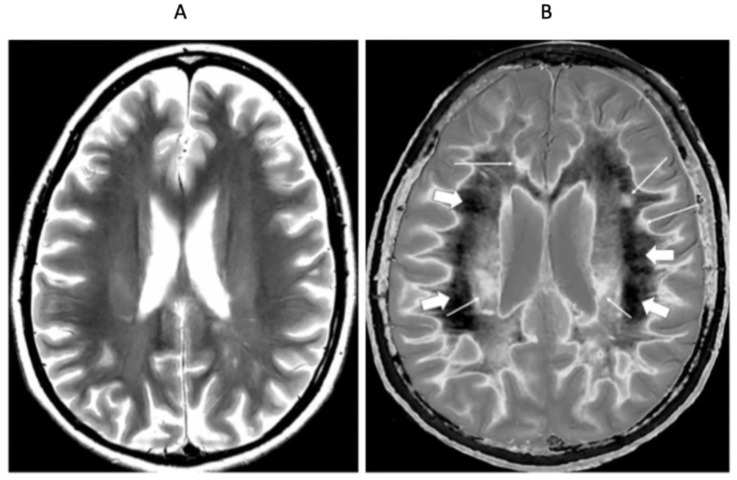
Forty-one-year-old female with MS in remission. Comparison of 2D T_2_-wSE (**A**) and narrow mD dSIR (**B**) images. No abnormality is seen on the T_2_-wSE image, but three focal lesions are seen on the dSIR image (long thin arrows). The corticospinal tracts are also seen (short thin arrows). The normal superior longitudinal fasciculi are of intermediate (posterior) to low (anterior) signal in (**B**). More peripheral white matter is normal and has a low signal in (**B**) (thick arrows). A high signal boundary is seen between white matter and cortical grey matter as well as between white matter and CSF around the lateral ventricles in (**B**).

**Figure 7 diagnostics-15-00329-f007:**
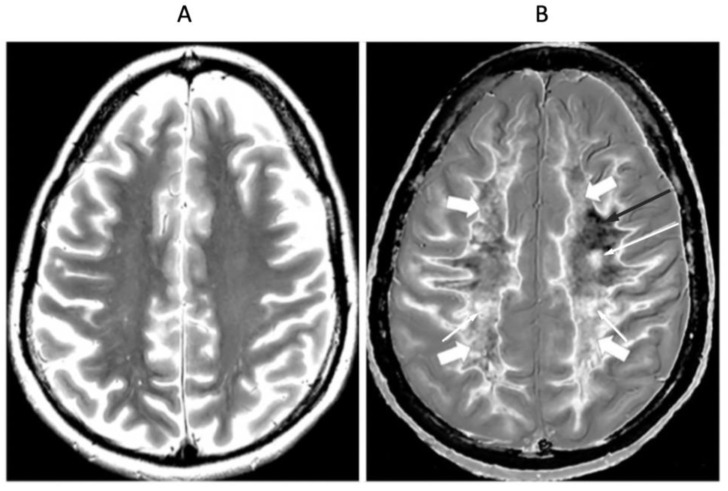
Forty-one-year-old female patient with MS in remission (as for Figure 6). Two-dimensional T_2_-wSE (**A**) and narrow mD dSIR (**B**) images at the same level. No abnormality is seen in (**A**). A focal lesion is seen in (**B**) (long thin white arrow) and the corticospinal tracts show a high signal (short thin white arrows). In addition, there is widespread, patchy increased signal in white matter (short thick white arrows) with only a small region showing a normal or near normal low signal (long black arrow). High contrast and high spatial resolution contrast are seen at the boundaries between normal white matter and normal gray matter in (**B**). These features are less obvious in areas where the white matter shows abnormal high signal.

**Figure 8 diagnostics-15-00329-f008:**
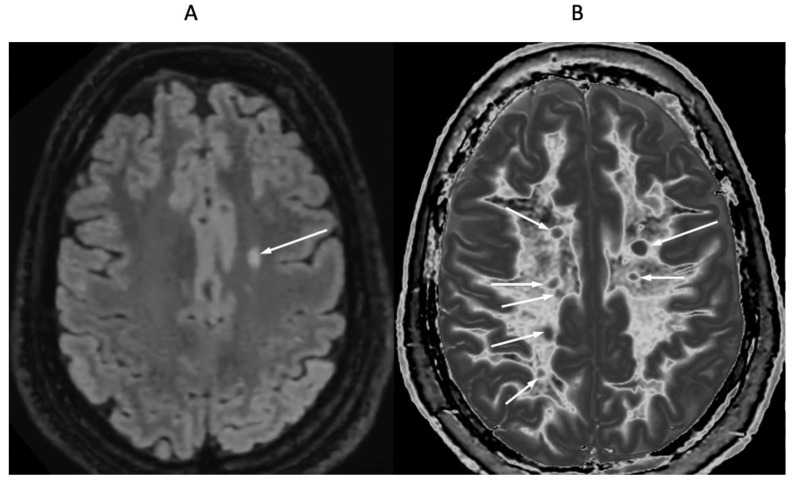
Thirty-two-year-old female patient with MS undergoing a relapse. T_2_-FLAIR (**A**) and synthetic narrow mD dSIR (**B**) images. On the T_2_-FLAIR image, one lesion is seen (long arrow). The surrounding white matter appears normal. On the dSIR image (**B**), the lesion shown on the T_2_-FLAIR image is seen (long arrow) as well as six other lesions (short arrows). Most of the white matter in (**B**) is high signal corresponding to a high grade 5 (out of 5) whiteout sign [7]. This is in contradistinction to the appearance of the peripheral white matter in the patient with MS in remission shown in Figure 6B. where the peripheral white matter is low signal.

**Figure 9 diagnostics-15-00329-f009:**
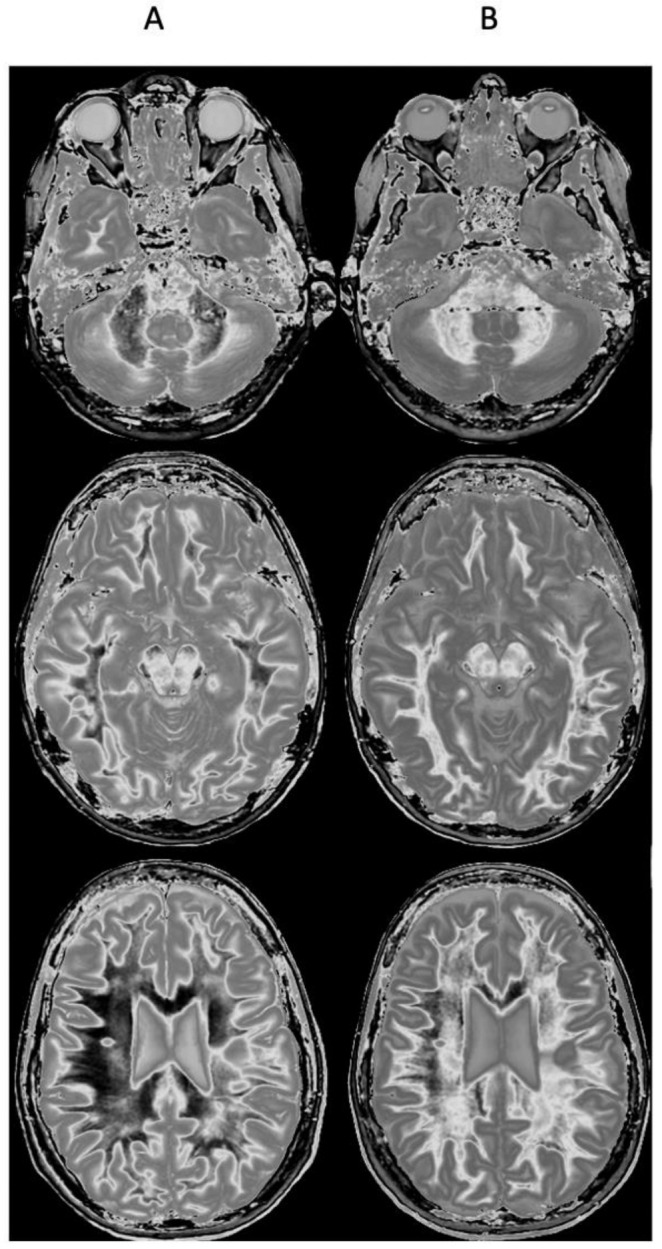
Thirty-eight-year-old female patient with MS in remission (**A**, left column) and during a relapse (**B**, right column) two years and five months later. Matching narrow mD dSIR images at three lower levels in the brain. In remission (left column), there are bilateral symmetrical areas of low signal in the white matter of the cerebellar and cerebral hemispheres. The corresponding areas show increased signal during the relapse (right column) consistent with a grade 4–5 (out of 5) whiteout sign. No evidence of a whiteout sign was seen on the T_2_-FLAIR images.

**Figure 10 diagnostics-15-00329-f010:**
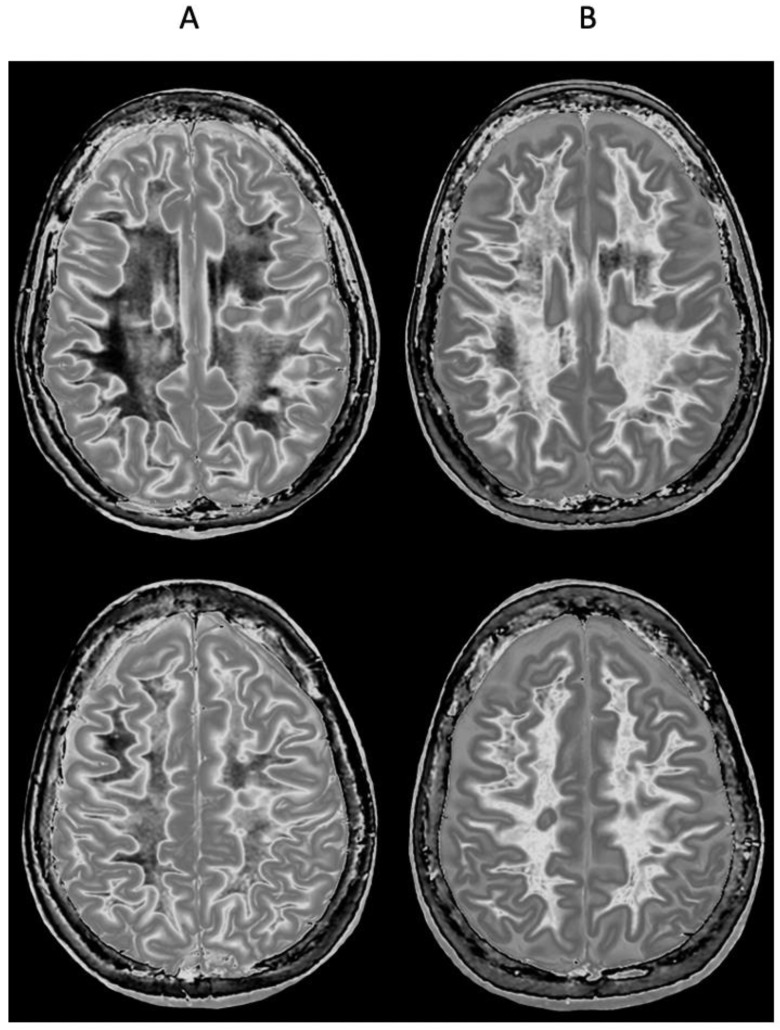
Thirty-eight-year-old female patient with MS in remission (**A**, left column) and during a relapse (**B**, right column) two years and five months later (same case as in Figure 9). Matching narrow mD dSIR images at two higher levels in the brain. In remission, white matter shows a low signal (left column). This is increased during the relapse in a bilateral symmetrical distribution consistent with a grade 4–5 (out of 5) whiteout sign (right column). No evidence of a whiteout sign was seen on the T_2_-FLAIR images.

**Figure 11 diagnostics-15-00329-f011:**
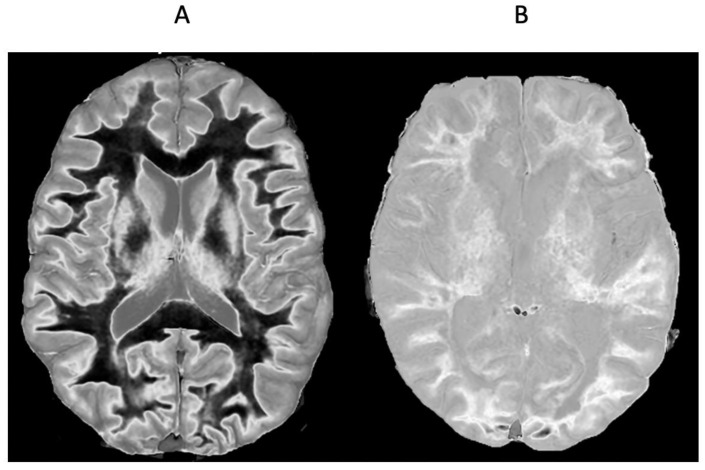
Skull-stripped images of a normal control (**A**) and 77-year-old patient with MS during a relapse (**B**). There is low signal in the normal white matter in (**A**) and a whiteout sign in (**B**). In addition, the gray matter in the thalamus and cortex has a uniform low signal and is nearly isointense with CSF. These are grayout signs. No evidence of a whiteout sign or grayout signs was seen on the positionally matched T_2_-FLAIR images.

**Figure 12 diagnostics-15-00329-f012:**
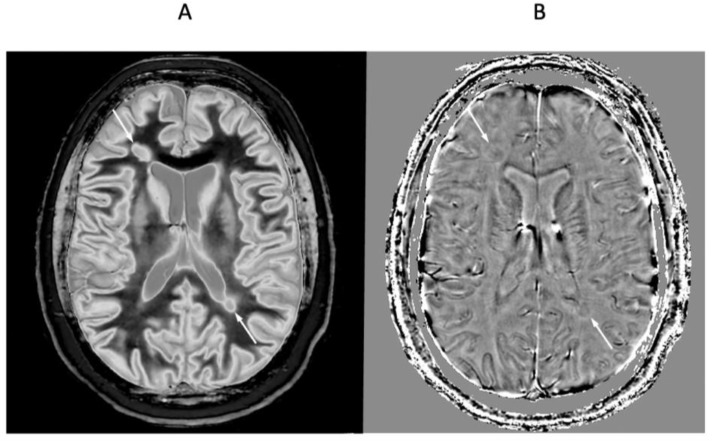
Forty-one-year-old female patient with MS during a relapse. Wide mD dSIR (**A**) and phase filtered susceptibility weighted gradient echo (**B**) images. The dSIR images show well defined high signal boundaries around two lesions in (**A**) (white arrows). These lesions show less obvious rim signs on the susceptibility weighted images (**B**) (white arrows).

**Figure 13 diagnostics-15-00329-f013:**
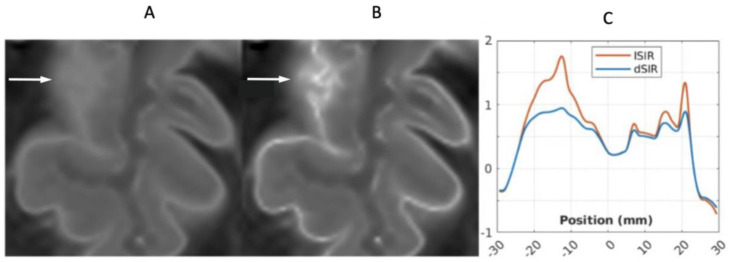
Forty-one-year-old female patient with MS. A leucocortical lesion is shown in the right medial frontal lobe on the narrow mD dSIR image (**A**) and the matching synthetic lSIR image (**B**) (arrows). There are also left to right profiles with signal plotted against position (in mm) for the dSIR (blue) and lSIR (orange) images (**C**) at the level of the horizontal arrows shown in (**A**,**B**). No boundary between white matter and gray matter is seen within the lesion in (**A**). A disrupted high signal boundary between white matter and gray matter is seen in the lesion in (**B**). The lSIR profile (orange) has higher signal and generally steeper slopes than the dSIR profile (blue) in (**C**). The difference in signal (or contrast) achieved for the same change in position is generally greater with the lSIR filter.

**Figure 14 diagnostics-15-00329-f014:**
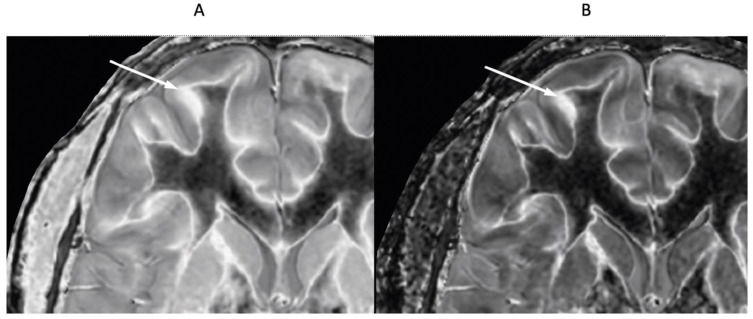
Twenty-four-year-old female patient with MS in remission (TI_s_ = 350 ms, TI_i_ = 500 ms). Narrow mD dSIR (**A**) and composite (T_1_ and T_2_) cdSIR (TI_s_/TE_s_ = 350/7 and 80 ms; TI_i_/TE_i_ = 500/7 ms) (**B**) images. A leucocortical lesion is seen on the dSIR image (**A**) and with higher contrast on the cdSIR image (**B**) (arrows). White matter and gray matter boundaries show higher contrast and are narrower on the composite filter cdSIR image (**B**). Also, white matter is more uniformly low signal in (**B**). These features are consistent with the attenuation of the first IR TI_s_ filter signal increasing contrast and narrowing boundaries at the positive pole and decreasing these at the negative pole as shown in Figure 5A.

**Figure 15 diagnostics-15-00329-f015:**
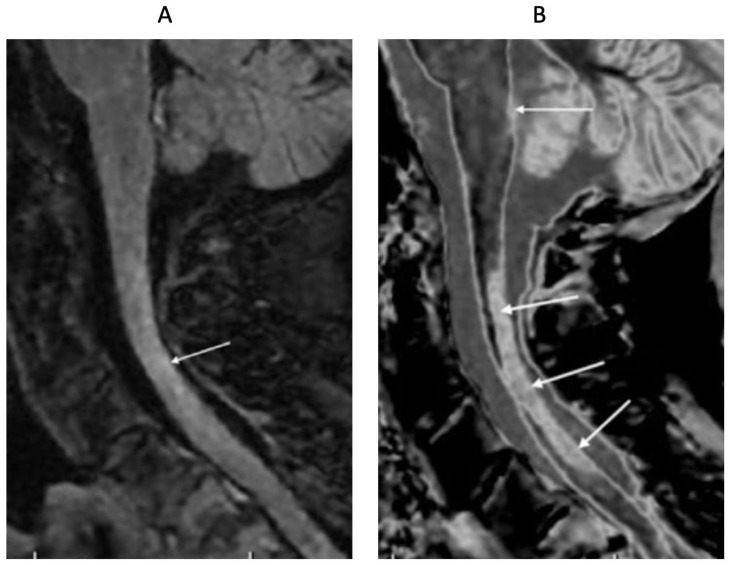
Forty-three-year-old female patient with MS in remission. Sagittal 3D T_2_-FLAIR (**A**) and 3D wide mD dSIR (**B**) images. The T_2_-FLAIR image shows a poorly defined area of increased signal in the cervical cord (arrow). The dSIR image shows a high contrast lesion with sharply defined boundaries, which is much more extensive than in (**A**) (lower three arrows). An additional lesion is seen in the medulla on the dSIR image (highest arrow) (**B**), but not on the T_2_-FLAIR image.

**Figure 16 diagnostics-15-00329-f016:**
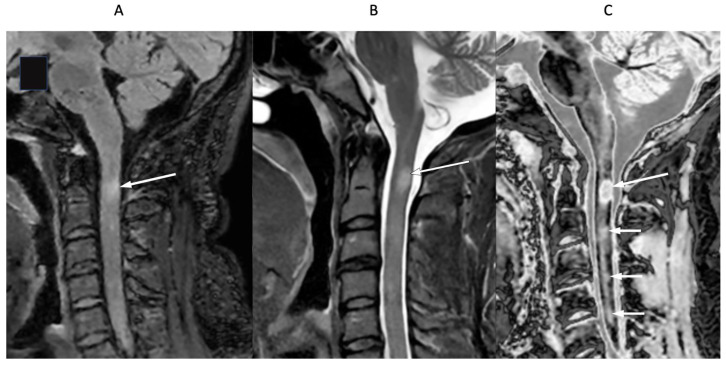
Forty-two-year-old female patient with MS in remission. Sagittal T_2_-FLAIR (**A**), T_2_-wSE (**B**), and wide mD dSIR (**C**) images. A large lesion is seen on all three images but with higher contrast on the dSIR image (long arrows). Three small lesions are only seen on the dSIR image (short arrows).

**Figure 17 diagnostics-15-00329-f017:**
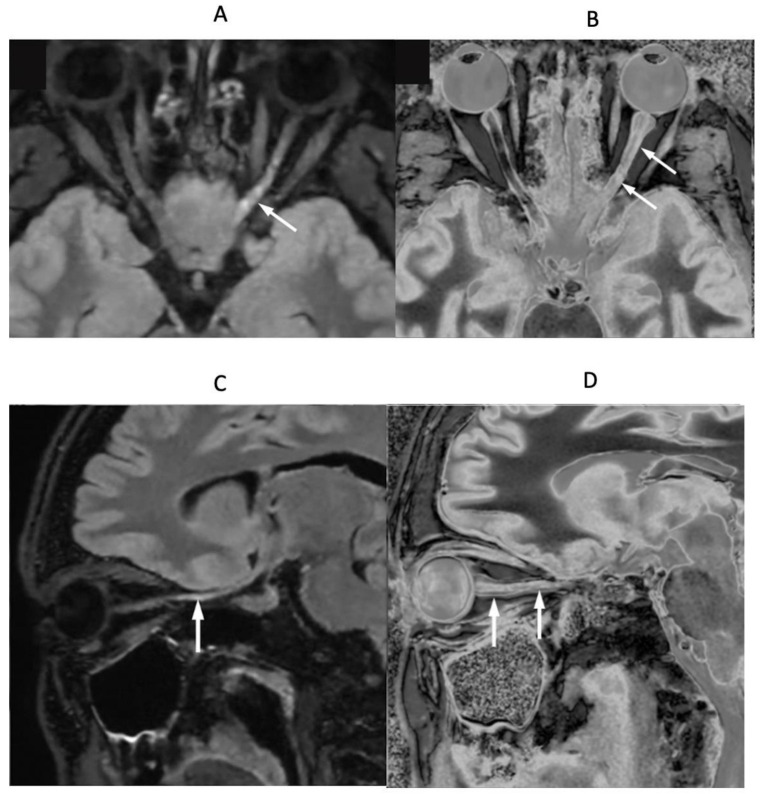
Patient with optic neuritis and suspected MS. The upper row shows axial fat saturated T_2_-FLAIR (**A**) and narrow mD dSIR (**B**) images. The lower row shows matched oblique fat saturated T_2_-FLAIR (**C**) and narrow mD dSIR (**D**) images through the left optic nerve. Distal changes are seen in the left optic nerve on the T_2_-FLAIR images (arrows) but both proximal and distal changes are seen in the left optic nerve on the dSIR images (arrows).

**Table 1 diagnostics-15-00329-t001:** Pulse sequences and pulse sequence parameters.

#	Sequence	TR (ms)	TI (ms)	TE (ms)	Matrix Size Voxel Sizes (mm)	Number of Slices	Slice Thickness (mm)
1	2D FSE IR (for white matter nulling)	9192	350	7	256 × 224	26	4
					0.9 × 0.1		
					Z512		
					0.4 × 0.4		
2	2D FSE IR (used with #1 for narrow mD dSIR)	5796	500	7	256 × 224	26	4
					0.9 × 0.1		
					Z512		
					0.4 × 0.4		
3	2D FSE IR (used with T_1_ for wide mD dSIR)	5796	800	7	256 × 224	26	4
					0.9 × 0.1		
					Z512		
					0.4 × 0.4		
4	3D BRAVO (for white matter nulling)	2000	400		256 × 256	220	0.8
					0.8 × 0.8		
					Z512		
5	3D BRAVO (used with #4 for wide mD dSIR)	2000	800		256 × 256	220	0.8
					0.8 × 0.8		
					Z512		
6	2D T_2_-FLAIR	6300	1851	102	320 × 240	26	4
					0.7 × 0.7		
					Z512		
					0.4 × 0.4		
7	3D T_2_-FLAIR without/with fat saturation	6300	1850	102	256 × 256	252	0.8
					0.8 × 0.8		
					Z512		
					0.6 × 0.6		
8	3D susceptibility weighted	40	-	32	300 × 300	110	2
					0.8 × 0.8		
					Z512		

Z = zipped.

**Table 2 diagnostics-15-00329-t002:** Bipolar filters.

Bipolar Filter	Reverse Bipolar Filter	Tissue Property
SIR	rSIR	T_1_
dSIR	drSIR	T1
cdSIR	cdrSIR	T1, T2, T2*, D*
lSIR	lrSIR	T1
clSIR	clrSIR	T1, T2, T2*, D*
Synthetic SIR, dSIR, lSIR	rSIR, drSIR, lrSIR	T1
Synthetic cdSIR, clSIR	cdrSIR, clrSIR	T1, T2, T2*, D*

**Table 3 diagnostics-15-00329-t003:** Bipolar filters signal equations and other functions.

#	Filter, Other Functions	Signal Equation	Figure #
1	IR, TI_s_	S_TIs_ = 1 − 2e^−TIs/T^_1_	Figure 1, Figure 2 and Figure 3
2	IR, TI_i_	S_TIi_ = 1 − 2e^−TIi/T^_1_	Figure 1, Figure 2 and Figure 3
3	SIR	S_SIR_ = S_TIs_ − S_TIi_	Figure 1
4	dSIR	$S_{\mathrm{dSIR}}=\frac{S_{\mathrm{TIs}}-{\;S}_{\mathrm{TIi}}}{S_{\mathrm{TIs}}+{+S}_{\mathrm{TIi}}}(\frac{\mathrm{difference}}{\mathrm{sum}})$	Figure 2 and Figure 3
5	cdSIR	$S_{\mathrm{dSIR}}=\frac{S_{\mathrm{TIs}}\cdot S_{\mathrm{OF}}-{\;S}_{\mathrm{TIi}}}{S_{\mathrm{TIs}}\cdot S_{\mathrm{OF}}+S_{\mathrm{TIi}}}$ $S_{\mathrm{dSIR}}=\frac{S_{\mathrm{TIs}}-{\;S}_{\mathrm{TIi}}\cdot S_{\mathrm{OF}}}{S_{\mathrm{TIs}}+S_{\mathrm{TIi}}\cdot S_{\mathrm{OF}}}$	Figure 5
6	cdSIR, S_OF_	S_OF_ = ±e^−ΔTE/T2^, ±e^−ΔTE/T2*^, ±e^−ΔbD*^, etc.	Figure 5
7	dSIR, S_dSIR_	$S_{\mathrm{dSIR}}\;\approx \;\frac{\ln\;4}{∆TI}\;\;T_{1}-\frac{\Sigma TI}{∆TI}$(in mD)	Figure 2
8	dSIR, T_1_	$T_{1}\approx \frac{∆TI}{ln4}\;S_{\mathrm{dSIR}}+\frac{\Sigma TI}{ln4}$(in mD)	Figure 2
9	lSIR	S_lSIR_ = ½(ln S_TIs_ − ln S_TIi_)	Figure 4
10	clSIR	$S_{1\mathrm{SIR}}{=\mathrm{lnS}}_{\mathrm{Tis}}\cdot S_{\mathrm{OF}}-{\mathrm{lnS}}_{\mathrm{TIi}}$ $S_{1\mathrm{SIR}}{=\mathrm{lnS}}_{\mathrm{Tis}}-{\mathrm{lnS}}_{\mathrm{TIi}}\cdot S_{\mathrm{OF}}$	Figure 4
11	clSIR, lSIR	$S_{c1\mathrm{SIR}}{=S}_{c1\mathrm{SIR}}\pm \frac{\Delta \mathrm{TE}}{T_{2}}$$,\;\pm \frac{\Delta \mathrm{TE}}{{T_{2}}^{*}}$, ±ΔbD*, etc	Figure 4
12	lSIR, dSIR ^†^	S_lSIR_ = atanh S_dSIR_	Figure 4

^†^ The lSIR filter is the inverse hyperbolic tangent of the dSIR filter.

## Data Availability

No new data were created or analyzed in this study. Data sharing is not applicable to this article.

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
