# Peer review of "Ultra-High Contrast (UHC) MRI of the Brain, Spinal Cord and Optic Nerves in Multiple Sclerosis Using Directly Acquired and Synthetic Bipolar Filter (BLAIR) Images"

_diagnostics, 2025, doi:10.3390/diagnostics15030329_

Round 1
Reviewer 1 Report
Comments and Suggestions for Authors
The manuscript, titled "Ultra-High Contrast (UHC) MRI of the Brain, Spinal Cord and Optic Nerve in Multiple Sclerosis Using Directly Acquired and Synthetic Bipolar Filter (BLAIR) Images", investigates a novel imaging approach utilizing ultra-high contrast (UHC) MRI techniques for diagnosing and understanding multiple sclerosis (MS). The study introduces and evaluates bipolar filter sequences, including divided subtracted inversion recovery (dSIR), logarithmic subtracted inversion recovery (lSIR), and composite divided subtracted inversion recovery (cdSIR). These methods enhance the visualization of abnormalities in the brain, spinal cord, and optic nerves of MS patients by increasing contrast in areas where conventional MRI techniques fail to detect significant changes.
Reasons for Major Revisions
While the manuscript introduces a compelling imaging innovation, several issues necessitate major revisions:
-
Abstract Improvement (Lines 20–33): The abstract lacks clarity and fails to succinctly convey the significance, methods, and results of the study. The authors should streamline the abstract by emphasizing the novelty of the bipolar filter technique and its clinical implications.
-
Overly Technical Introduction (Lines 38–72): The introduction includes dense technical jargon without adequately establishing the broader clinical relevance of the study. Simplify the language and clearly state the problem and the research question.
-
Figures and Legends (e.g., Lines 105–118): The descriptions of Figures 1 and 2 are highly technical. Authors should simplify and clarify figure legends to make them more comprehensible to a broader audience.
-
Methods Section Clarity (Lines 289–303): While the study employs innovative MRI techniques, the methods lack sufficient detail for reproducibility. Include more information about patient selection criteria and image analysis protocols.
-
Inadequate Discussion of Clinical Implications (Lines 463–531): The discussion focuses heavily on technical aspects but does not adequately relate findings to clinical practice. Elaborate on how bipolar filter MRI could change current diagnostic or treatment paradigms for multiple sclerosis.
-
Formatting of Equations and Variables (Lines 78–100, 187–200): The equations are presented without enough explanatory text. Clarify the meaning of symbols and parameters for readers who may not have a physics background.
-
Abstract and Keywords Redundancy (Lines 34–36): The keywords repeat terms already mentioned in the abstract (e.g., “whiteout sign”). Revise to include more diverse and specific terms that enhance discoverability.
-
Conclusion Relevance (Lines 590–594): The conclusion is overly general and fails to emphasize key findings or actionable insights. The authors should focus on how their results advance the field and propose future research directions.
Author Response
Thank you for your thoughtful and detailed review and the encouragement to extend the work beyond the scope of the original submission.
Comment 1: While the manuscript introduces a compelling imaging innovation, several issues necessitate major revisions ..
Response 1: Thank you for the overall comment that the manuscript introduces a compelling imaging innovation and your direction on revisions.
Comment 2: The authors should streamline the abstract by emphasizing the novelty of the bipolar filter technique and its clinical implications.
Response 2: Thank you also for this positive comment. We have introduced some text about the novelty of the bipolar filter and an additional comment about its clinical implications.
Comment 3: The introduction includes dense technical jargon without adequately establishing the broader clinical relevance of the study. Simplify the language and clearly state the problem and the research question.
Response 3: We agree that the text is dense in places, but it is difficult to describe a new technique without introducing some new concepts and we have tried to keep it brief. We also agree that stating the problem and research question is a good idea for a research paper, but this is an educational review with educational objectives rather than answers to a research question.
Comment 4: The descriptions of Figures 1 and 2 are highly technical. Authors should simplify and clarify figure legends to make them more comprehensible to a broader audience.
Response 4: We also agree that the details of Figs. 1 and 2 are technical but this an attempt to present in visual form the information conveyed in the equations in Table 3 (lines 1-4). The concepts are new but we have used elementary differential calculus as the basic method of explanation and think it would be very difficult to get a simpler but coherent explanation of why the contrast achieved with dSIR sequences can be ten or more times higher than that achieved with conventional IR sequences.
Comments 5: While the study employs innovative MRI techniques, the methods lack sufficient detail for reproducibility. Include more information about patient selection criteria and image analysis protocols.
Response 5: We have included TR, TI, TE, matrix size, slice thickness etc in Table 1 for each of the sequences used. This has been sufficient to implement the techniques on GE, Philips and Siemens MR systems. The techniques use existing sequences that are already on the system and do not require very detailed explanation which would be the case with implementing entirely new sequences de novo. The sequence parameters essential for determining contrast are included, and this seems appropriate for image interpretation purposes. The patients were clinically assessed and graded for this educational purpose into relapse and remission groups. In terms of image analysis, grading of the whiteout sign is described in a referenced paper [7].
Comment 6: The discussion focuses heavily on technical aspects but does not adequately relate findings to clinical practice. Elaborate on how bipolar filter MRI could change current diagnostic or treatment paradigms for multiple sclerosis.
Response 6: Thank you for the generous offer to elaborate on how bipolar filter MRI could change diagnostic or treatment paradigms for multiple sclerosis. This is a major topic which has been evolving for 40 years and may be updated when the 5th edition of the McDonald criteria are published later this year. To be honest, we think it is probably premature to speculate on how diagnosis of MS may be changed by BLAIR sequences based on a small series of illustrative cases but there are key findings such as the presence of the whiteout sign during relapses that may be transformative. The whiteout sign and better understanding of boundaries/rims of MS lesions may improve resolution of the clinical radiological paradox whereby patients have symptoms of a relapse but show stable appearances on MRI.
Comment 7: The equations are presented without enough explanatory text. Clarify the meaning of symbols and parameters for readers who may not have a physics background.
Response 7: We have expanded the definitions of the terms used in the equations in the text in each case.
Comment 8: The keywords repeat terms already mentioned in the abstract (e.g., “whiteout sign”). Revise to include more diverse and specific terms that enhance discoverability.
Response 8: Yes, some of the key words are included in the abstract, but the abstract and keywords serve different purposes so we hope that this is acceptable. We have expanded the keywords to include white matter disease of the brain.
Comment 9: The conclusion is overly general and fails to emphasize key findings or actionable insights. The authors should focus on how their results advance the field and propose future research directions.
Response 9: Thanks for the invitation to expand on our findings and elaborate on future clinical and research directions. We have mentioned further technical development and have added clinical evaluation to this, but have not speculated too far beyond the available evidence.
Reviewer 2 Report
Comments and Suggestions for Authors
The paper is an insightful review of ultra-high contrast bipolar MRI techniques, which offers a new perspective on multiple sclerosis. The bipolar techniques could be combined with various acquisition sequences. However, the present manuscript shows some minor language errors, but the paper is still readable.
Because the ultra-high contrast bipolar filter is a rather new contrast, please discuss if the bipolar technique could help to identify neurological conditions that show negative findings in DWI,
T1, T2 MRI. In particular, the preventive medicine department would be interested in transient ischemic stroke, migraine, and other headaches.
Author Response
Comment 1: The paper is an insightful review of ultra-high contrast bipolar MRI techniques, which offers a new perspective on multiple sclerosis.
Response to comment 1: Thank you very much for your very favorable comment.
Comment 2: However, the present manuscript shows some minor language errors, but the paper is still readable.
Response to comment 2: We have checked the manuscript and improved the language.
Comment 3: Because the ultra-high contrast bipolar filter is a rather new contrast, please discuss if the bipolar technique could help to identify neurological conditions that show negative findings in DWI,
T1, T2 MRI. In particular, the preventive medicine department would be interested in transient ischemic stroke, migraine, and other headaches.
Response to comment 3: Thank you for the invitation to generalize the results from MS to include other conditions. So far we have limited experience in TIA, migraine and other headaches but think these conditions are well worth further study. There may be small changes in T1, T2 and/or D* which could be detected with TP-BLAIR sequences and would be of clinical importance.
Round 2
Reviewer 1 Report
Comments and Suggestions for Authors
Thanks for addressing the comments.